

**Integration of Remote Sense and Geographic Information Systems in Geological Faults**
**Detection in Crete Island, Greece**.
**Mohamed Elhag[1*], Dalal Alshamsi[2]**
[1] Department of Hydrology and Water Resources Management, Faculty of Meteorology,
Environment & Arid Land Agriculture, King Abdulaziz University Jeddah, 21589. Saudi Arabia.
[2] Department of Geology, College of Science, United Arab Emirates University.
*Correspondence to: melhag@kau.edu.sa*
**Abstract**
Fracture systems are of great importance in the field of structural geology. Faults commonly
afford easy passage to groundwater and fluids such as hydrothermal fluids and magmas (mineral
entrapment over the years) or even contribute in earthquake hazard monitoring. For a geologist it
is not always easy to discern such morphotectonic structures at close range (i.e. heavy
overgrowth of vegetation). Both remote sensing techniques and spatial modeling (GIS) permit
the recognition and better understanding of the brittle tectonics in an area. This study was an
effort to delineate the tectonic structures (i.e. fault system) on the Crete Islands by combining
Sentienl-2 satellite data and spatial data. For the enhancement and better discrimination of
photolineaments primarily recognized on satellite imagery, a variety of enhancement techniques
have been applied. The evaluation of a photolineament as a potential fracture zone was based on
several factors; the DEM of the study area, the shaded relief, the slopes and corresponding
aspects, the drainage network, the geology and general observations on vegetative coverage
appearance. The application of these methods revealed several fracture zones, which we
recommend being certified by field investigations. Fault-mapping results may be used for a
variety of purposes. Indicative places of large concentration of groundwater are of vital
importance for subsequent exploitation by areas of need. Furthermore, because the well-known
Anatolian fault zone extends over the Northern part of Crete, the present work may provide
useful information for further analysis by geophysists and seismologists.
**Keywords:** DEM. Fracture Detection, Morphometric, Morphotectonic, Remote Sensing,
Photolineaments.



**1. Introduction**
O' Leary et al. (1976) and Colwell, (1983) gave a good-informing definition of the term
lineament. It refers to a mappable, single or composite linear feature of a surface, whose parts are
aligned in a rectilinear or slightly curvilinear relationship and which differs distinctly from the
pattern of adjacent features and presumably reflects a subsurface phenomenon.
An extensive literature review on the consideration of linear features and lineaments is well
presented by Siegal and Gillespie (1980). Suffice to say here, that lineaments identified on aerial
photographs and satellite images may represent diverse topographic features (drainage lines),
vegetation/soil alignments, coastal lines, crests, ridges, stratigraphic contacts, fold axis (foliation)
and seismic zones (Lunkka 1994).
The importance of detecting lineaments lies in the fact that they often represent fault systems.
Faults indicate failure of the crust along a surface, accompanied by the relative movement of the
geological units from both sides of that surface (Caumon et al. 2009). Such a zone of structural
weakness is a major component in structural geology and may be related to a series of other
phenomena. It is noticeable that, in many cases in the past, such a system has been related to
seismic events occurring in an area. Locating and identifying the movement is of dual
importance to a geologist as the impacts of seismic hazards may encompass of both lives and
economic losses (Colwell 1983).
Furthermore, faults identification is of economic importance too. That is for; they commonly
afford easy passage to several fluids like water, hydrothermal fluids and magmas (Oelkers et al.
2009). Groundwater on the one hand, can be easily transferred through such channel ways,
sometimes over large distances and finally reaching and supplying areas of need. On the other



hand, mineral entrapping at some time in the past may be of advantageous exploration use (Elhag
and Bahrawi 2014).
In the case of faults detection, integration of Remote Sensing and Geographic Information
Systems is an issue of great interest. That is for, photolineaments on satellite images do not
always account for failure of the crust. They may well represent the drainage lines of the area
(rivers), geological boundaries of formations or even cultural features such as roads. The criteria
for faults investigation are discussed in several aspects (Greenbaum 1992; Schulson 2004).
Vegetation may play a great role in photogeology for it may reveal underlying structural features
not easily discerned at close range. As far as fracturing is concerned, attention should be given to
preferential growth of vegetation along linear-curvilinear surfaces (Rodomsky 2011). The reason
for this, as mentioned earlier on, is that fractures often act as channel for underground water.
Water subsequently, increases the moisture content of soil (in relation to surrounding area) and
encourages in one sense, a preferential alignment of vegetation along these fracture zones
(Singhal and Gupta 2010). Moreover, it is not rare to identify on an image (or photograph) abrupt
changes in types of vegetation or even sudden disappearance of a certain plant species.
Particularly in the case where vegetation varies over a surface underlain by the same type of
bedrock, fracturing is implied (Phillips et al. 2008).
Faults in some cases, may even throw permeable against impermeable rocks. Water again finds
ways along the zone of structural weakness and reaches the Earth's surface by numerous springs
(along the contact of the two formations). At these points, high moisture conditions are
particularly favorable to intense vegetation growth (Singhal and Gupta 2010).
Abrupt changes in slope are commonly associated with brittle tectonics (Agliardi et al. 2001;
Agliardi et al. 2009). Attention should be given though to the orientation of slopes with respect
to the illumination source. It may have an impact on vegetation and hence confuse the
photointerpreter. As Singhal et al. (2010) pointed out, this factor may greatly influence the
survival of plants and explained that moisture content along the surface of a given slope varies.
That is for; the parts that are not so well exposed to sun lighting retain less moisture than the
more illuminated ones and hence a contrast of low-to-high density vegetation growth occurs.
The knowledge of underlying bedrock type is crucial for the processing of a photogeologist's
work. Any truncation and displacement of beds may effectively reveal fracturing (Odling et al.
1999). Abrupt changes in geological formations may also indicate fault zones (Stein 1991). A
fault may bring in contact rocks of different petrology and general characteristics that show no
physical or inherent association between the two formations. In other words, these formations are
not expected to be in contact with each other unless fracturing has occurred at some time in the
past (Boggs 2009).
Variations in vegetation discussed above, may be also due to petrological differences of
underlying rocks, without necessarily the presence of a fault (Rodomsky 2011). One explanation
for such variations is that of bedrock influencing soil composition and consequently the plant
species that can exist.
This study was an effort to delineate the brittle tectonics of Crete islands in Kolymvari area
(Greece) by using spatial models and digital image processing. The satellite image that was
initially provided in the laboratory was enhanced by a variety of methods for the better
discrimination of photolineaments. Certainly, not all lineaments detected, were expected to
represent failure of the crust (faults) and could be easily confused with roads and rivers. They



had to be studied with respect to real-life conditions. In particular, information related to
geomorphology (elevation, slopes, aspect, and drainage system) and geology of the region (types
of rocks, boundaries of geological formations) was integrated with the satellite data and the final
evaluation of a certain lineament, as a potential fault was carried out.
**2. Materials and Methods**
**2.1. Study Area**
The hydrological basin of the study area is situated in the western part of Crete Island and is
referred to Municipality of Kolymvari (Figure 1). The landscape structure is fundamentally
mountainous, resulting in plain only near the coast. The area has a sub-humid Mediterranean
climate with an annual average temperature of 19.96°C (Elhag and Bahrawi 2016). The
watershed of Kolymvari is mainly an agricultural area where the most common cultivations are
olive trees, citrus trees, vineyards, and vegetables. The area has also light industrial activities
such as olive mills, wineries, and other agricultural factories. In the coastal zone of Keritis
watershed, there are many touristic units (Papafilippaki et al. 2007). However, the possible flow
paths were all directed to Kolymvari Stream, which is, therefore, the surface water body (Elhag
et al. 2017). From geotectonic point of view, it has not been easy to classify. Lack of fossils in
many cases and difficulties in comprehending the exact geo-processes involved in the formation
of the Aegean Sea during the Pliocene and Pleistocene, have been an issue under investigation
from many geologists over the past years.
**2.2. Remote Sensing Data**
The goal of digital image processing was to improve such spectral responses and generate
images more interpretable than the original ones, where photolineaments would be better



discerned. The digital image of Sentienl-2A (Tile Number: T34SGE, with 0% cloud cover)
dating 12/03/2018 and covering the pilot area, has been subject to several enhancement
techniques. There were two basic types of distortions that had to be reckoned with, prior to any
of the enhancement methods; radiometric distortions and geometric distortions.

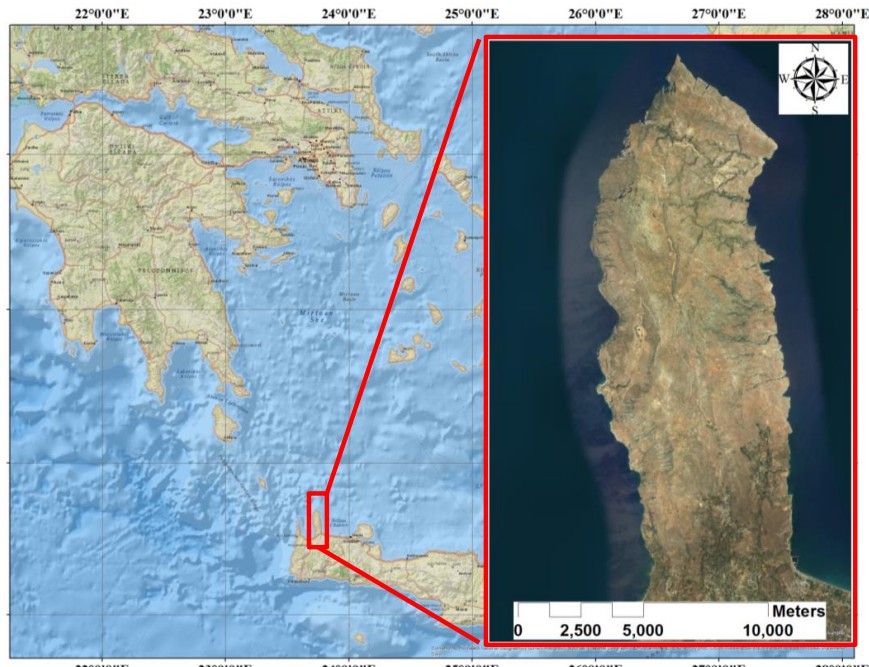

**Figure 1. Location of the study area**
**2.2.1. Lineaments Evaluation by RS means**
A variety of enhancement methods have been applied to digital raw data in order to improve the
spectral characteristics of objects and emphasize the photolineaments of the area under
investigation. The original bands have been contrast-stretched for DNs occupy more gray levels
than before, in accordance with their frequency of occurrence (histogram equalize stretching).
According to Oskoei and Huosheng (2010), the Isotropic Laplacian, non-directional convolution





filtering as well as edge-enhancement technique, have been applied to Sentienl-2 (for vegetation
and geological features are particularly responsive in these two wavelength bands). Principal
Component Analysis aimed to compress information in fewer bands than the original ones,
uncorrelated to each other. Images generated by rationing, accounted for the real spectral
characteristics and physical properties of objects (topography and illumination effects on the
brightness values of pixels, have been eliminated). Schematic flow chart of the implemented
procedure in digital image analysis is illustrated in Figure 2.

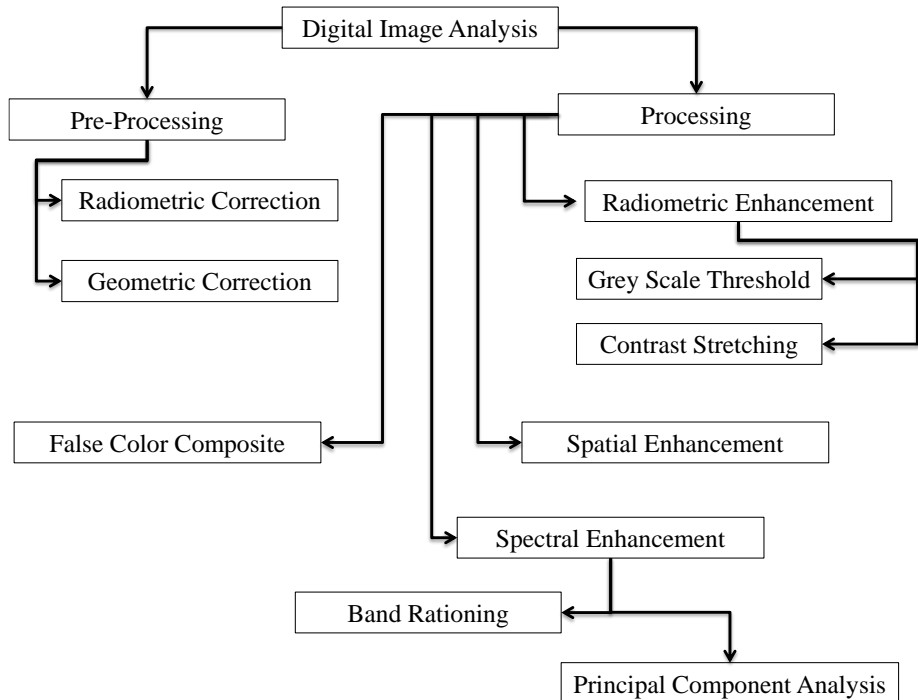


**Figure 2. Workflow of Digital Image Analysis**

**Grey level threshold and Contrast stretching**, Sentienl-2, B5 has turned out to be the most
appropriate one, for in this near infrared channel, land areas are highly reflected whereas water
areas are highly absorbed (Elhag 2017). In 16-bit computer encoding, a digital image can be



displayed over a dynamic range of 65536 gray levels (Lillesand et al. 2014). In many cases
however, only a small portion of these 65536 levels that are available by devices is utilized. The
radiometric ranges (minimum and maximum values of DN values) in each of the six (6) utilized
bands (Sentienl-2 B2-B7) of the original dataset are corrected.
**Spatial Enhancement,** the concept is based on a moving window (the so-called Kernel window)
that in short, contains an array of weighting factors, moves successively over all the pixels of a
single black and white image (individual band) and ascribes new weighted DN-values as a result
of the weighted original ones (Gupta 2017). A high pass filter, in order to emphasize high
frequency features or else local spatial detail (Aiazzi et al. 2002). The filter size was 5*5 and was
added back to the fourth band of the original band so that low frequency brightness information
was not totally lost.
**Principal Component Analysis** (PCA) is a unique mathematical transformation, designated to
reduce such redundancy in multispectral data (Kaarna et al. 2000). The idea is to compress all the
information contained in the n-original bands, into fewer than N-new channels/components
(Lillesand et al. 2014).
The high interband correlations (very close to 1) suggested that the best way to proceed with this
work was by applying a Principal Component Analysis (Table 1). No pair of bands presented
covariance =0 i.e. no pair of bands were completely independent to each other. Moreover, as
covariance > 0, data appeared to be positively correlated (Elhag 2016).
**Rationing**, it involves the division as well as more complex functions (additions, subtractions,
multiplications) between the DNs of two single bands (Wu et al. 2008). The technique is
indicative for both preserving the spectral reflectance characteristics of surficial matter and



masking brightness variations derived from illumination conditions and topographic effects
(Soulakellis et al. 2006).
**Table 1. Interband correlations.**

|  | Band 2 | Band 3 | Band 4 | Band 5 | Band 6 | Band 7 |
|---|---|---|---|---|---|---|
| **Band 2** | 1 | 0,950535 | 0,927525 | 0,244727 | 0,6878 | 0,767209418 |
| **Band 3** | 0,950535 | 1 | 0,974041 | 0,406527 | 0,7474 | 0,815756429 |
| **Band 4** | 0,927525 | 0,974041 | 1 | 0,418655 | 0,8131 | 0,879577936 |
| **Band 5** | 0,244727 | 0,406527 | 0,418655 | 1 | 0,607 | 0,468485418 |
| **Band 6** | 0,687812 | 0,7474 | 0,813061 | 0,607007 | 1 | 0,956571297 |
| **Band 7** | 0,767209 | 0,815756 | 0,879578 | 0,468485 | 0,9566 | 1 |


**False Color Composite (FCC),** one band at a time can be displayed in each of the three color-
guns; generating false - color composite images (objects do not appear in their natural colors)
that are undoubtedly more interpretable to human's eye than black and white images. For the
present work several combinations have been tried between original bands, stretched ones, the
principal component products as well as the ratio images (Pohl and Van Genderen 1998). The
scope was to produce a FCC image where Photolineaments would be discriminated to a
satisfactory degree for their further digitizing on screen. Table 2 summaries the used band
combinations and its uses.
**2.2.2. Lineaments Evaluation by GIS means**
Basically, not all photolineaments detected using Remote Sensing techniques are represented as
fracturing (Karnieli et al. 1996). Cultural features such as roads had to be readily recognizable on
satellite data to avoid their confusion with faults and lead to false photointerpretation results.
Schematic flow chart of the implemented procedure in spatial image analysis is illustrated in
Figure 3.



**Table 2. False Color Combinations**

| | **Band Combination** | **Uses** |
|---|---|---|
| **FCC1** | B4, B3, B2 | Vegetation outcrop in false color composite image appeared red. In cases of preferential vegetation appearance along linear/ curve linear features, potential underlying fracturing. |
| **FCC2** | PC1, B4, B2 | Geology was not easily discriminated whereas drainage network appeared partially enhanced (bluish tones). |
| **FCC3** | PC1, B6, B5 | NDVI's contribution in the green color-gun was reflected by the very bright green pixels. No other noteworthy remark could be made. Topography was totally lost. |
| **FCC4** | PC1, B4, PC2 | Vegetation appeared in bluish tones. The drainage lines in the northern part of the study area were well detected. |
| **FCC5** | PC1, B5convolved, B3 | No clear discrimination of the geology of the area was achieved. |
| **FCC6** | PC1, B5convolved, B7stretched | Less certain discrimination of the geological boundaries/geology of the area. No actual discrimination was made. |
| **FCC7** | PC1, PC2, PC3 | no prediction of the color characterizing each feature was easily made. The overthrust in the northern part was readily differentiated from the rest of the study area. |
| **FCC8** | PC1, PC2, B3 | The use of two PCs did not give noteworthy results. Bright green pixels represented vegetative coverage, whereas dark tones of green delineated to a limited extend the drainage network. The elongated feature was not enhanced in this product. |
| **FCC9** | PC1, B5convolved, PC3 | Topography was better expressed than in the previous case (PC1 in combination with PC2). Several lineaments were readily seen (for non-directional filtering of B5 component). |
| **FCC10** | B7enhanced, B5convolved, B3 | No particular enhancement was observed compared to the previous products. |
| **FCC11** | PC1, B5convolved, B5 | Several photolineaments were enhanced, topography was well expressed while stratigraphic contacts were not well delineated. |
| **FCC12** | PC1, B5convolved, B3 | Partial enhancement of drainage network and preferential alignments of vegetation (greenish tones). |
| **FCC13** | PC1, B6, B3 | Topographic sense was lost. Some vegetation alignments and abrupt changes in green tonal variations. |
| **FCC14** | PC1, B5, B3 | topography was very well expressed whereas objects appeared in very natural colors and tones. The drainage network was well delineated, and several lineaments were detected. The same difficulty met in previous combinations in discriminating between the diverse geological formations. |



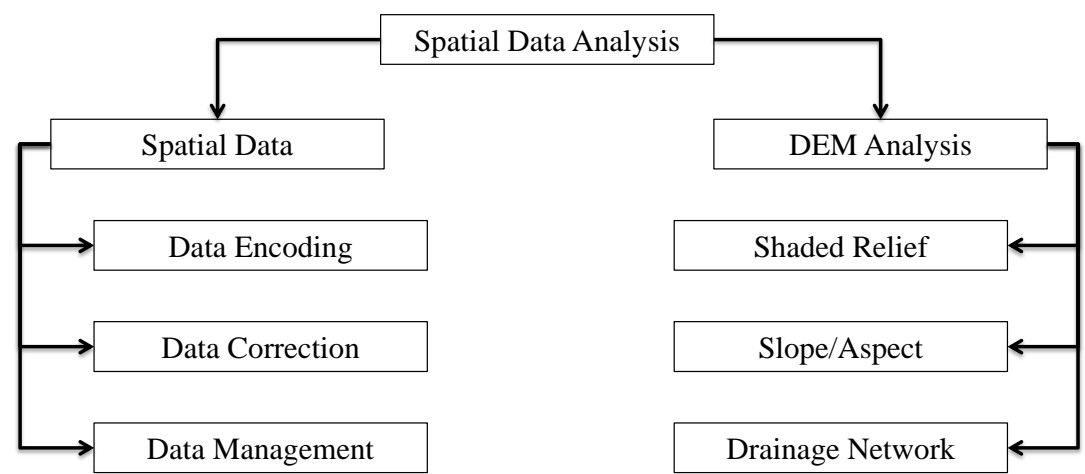


**Figure 3. Workflow of Spatial Image Analysis**

**3. Results and Discussion**
**3.1. RS Lineaments Evaluation**
To transform the original data onto the new PC axes, transformation coefficients (eigenvalues,
eigenvectors) should be obtained (Johnson and Wichern 2002). Eigenvalues are presented in
Table 3 and describe the variation within the dataset. The % of total scene variance explained by
each PC is given by:
(eigenvalue for PCx * 100) / (sum of all eigenvalues)
**Table 3. Eigenvalues - % of total variance**

| PC | Eigenvalue | % of Variance | Cumulative% of Variance |
|---|---|---|---|
| PC1 | 900.7121015 | 82.27270992 | 82.27270992 |
| PC2 | 124.3086818 | 11.354585 | 93.62729492 |
| PC3 | 58.20089834 | 5.316177743 | 98.94347266 |
| PC4 | 7.348228513 | 0.671200789 | 99.61467345 |
| PC5 | 3.339351294 | 0.305022526 | 99.91969598 |
| PC6 | 0.879159159 | 0.080304024 | 100 |
| SUM | 1094.788421 | | |



Where x :1,2,3 ... 6
PC1 contains the largest amount of total scene-variance (82, 27%) and hence, is the most
correlated component with the original bands. PC2 on the other hand, accounts for a smaller
amount of the remaining information (11,35%), PC3 for 5,3% and so on. The three first PCs
account for 98, 94% of total scene-variance. As Nikolakopoulos et al. (2008), pointed out, noise
is suppressed to the less correlated extracted PCs (Lukáš et al. 2006). Hence, the rest of the
principal components (PC4, PC5, PC6) have been ignored.
Eigenvector matrix (Table 4) provides 'us with the loadings or else relative contributions, of each
of the bands to each of the PCs.
**Table 4. Relative contribution of each band to each extracted PC.**

|            | PC1      | PC2      | PC3      | PC4      | PC5     | PC6          |
|------------|----------|----------|----------|----------|---------|--------------|
| **Band-2** | 0.256745 | -0.43474 | -0.45772 | 0.548567 | -0.439  | -0.205610634 |
| **Band-3** | 0. 179081| -0.19287 | -0.33306 | -0.06943 | 0.1385  | 0.892072198  |
| **Band-4** | 0.314162 | -0.29929 | -0.39863 | -0.407   | 0.5738  | -0.397391284 |
| **Band-5** | 0.259551 | 0.80314  | -0.51111 | -0.0477  | -0.147  | -0.050194624 |
| **Band-6** | 0.761122 | 0.122543 | 0.471784 | 0.346882 | 0.2476  | 0.038391621  |
| **Band-7** | 0.395779 | -0.1555  | 0.19195  | -0.63718 | -0.613  | 0.004267441  |


PC1 has only positive loadings. That means that in the output image no particular feature or
structures are expected to be enhanced. It seems that it is a little bit more correlated to Sentinel-2,
B7 in comparison with the rest of the bands. PC1 however, bears many similarities to all six
bands and can be used as an image of high quality on its own. It is a weighted product of more or
less all input images and reflects in a very good way the albedo and topography of the area.
PC2 is a contrast between the high positive loading of Sentinel-2, B5 and the negative loading of
Sentinel-2, B2. Hence, PC2 as a single black and white image emphasizes the different sets of



features that are sensitive to these bands. More particularly, the spectral responses of the
vegetation biomass that is present in the scene (Sentinel-2, B5) contrasted to the soil-vegetation
differentiations (Sentinel-2, B2).
As far as PC3 is concerned, the output image is of less significance, as no clear discrimination is
available.  Information related to the correlation between the bands and the extracted PC's are
also given in Table 5. Factor loadings R, were computed in Excel by using the following
formula:
R (xp) = eigenvector (xp) * eigenvalue (p) % / variance (x) 1/2
where x refers to the xth channel and p to the pth component (Johnson and Wichern 2002).
**Table 5. Factor loadings**

|          | PC1      | PC2       | PC3       |
|----------|----------|-----------|-----------|
| **Band-2** | 0.778556 | -0.48975  | -0.35283  |
| **Band-3** | 0.841781 | -0.33679  | -0.39796  |
| **Band-4** | 0.891965 | -0.31568  | -0.2877   |
| **Band-5** | 0.623355 | 0.716576  | -0.31203  |
| **Band-6** | 0.985093 | 0.058921  | 0. 155217 |
| **Band-7** | 0.968296 | -0.14133  | 0. 119375 |


The correlation coefficient R extends in a range of -1 to 1. This is nothing more than unit-
normalization so that loadings are between -1 and 1 (Johnson and Wichern 2002). The closer the
coefficient is to -1 or 1, the more significant is the contribution of a channel to the corresponding
PC. Similarly, a loading close to zero is an indication of practically no contribution of a band to
the corresponding PC. Thus, PC1 is highly correlated to all bands but mostly to Sentinel-2, B6
and Sentinel-2, B7. PC2 on the other hand, is highly correlated to Sentinel-2, B5 and may be
used instead of the latter for a color-composite image.





For vegetation is highly reflected in near infrared and low reflected in visible red, (Sentinel-2,
B4)/ (Sentinel-2, B5) ratio generated an image where vegetation appeared in dark tones. In an
analogous way, due to the comparatively high reflectance of vegetation in visible green to the
lower reflectance in mid infrared, vegetation in (Sentinel-2, B3)/ (Sentinel-2, B6) image
appeared in light tones. In both ratios, topography has been eliminated and the area gave the
impression of being flat. The two images however, as well as the NDVI product can be safely
viewed together for the detection of vegetation biomass. Positive NDVI values, correlate with
vegetation while null values correspond to rocks and soil (Johnson and Wichern 2002).
(Sentinel-2, B7) + (Sentinel-2, B4) ratio, was based on an idea to enhance stratigraphic contacts.
That is for Sentinel-2, B7 is important in geology for the discrimination of geologic rock type
whilst one of Sentinel-2, B4's purposes is the detection of geological boundaries where no
particular enhancement was noticeable though.
Efforts to use others than Sentinel-2, B2 band in the ratio applied resulted in negative pixel
values (black pixels within the study area) and hence, could not be used for further analysis. In
comparison with the rest of the ratios, $((Sentinel\text{-}2, B2)^2 + (Sentinel\text{-}2, B5)^2)^{1/2}$ product
delineated in a better way vegetation amount while several lineaments were emphasized.
Edge enhancement methods resulted in the enhancement of individual bands (black and white
images) to a satisfactory degree. However, due to the low capability of human's eye to discern
the slight spectral responses of objects in gray scaling, the need for multispectral imagery
(display of more than one bands at a time) resulted in the creation of False Color Composite
products (FCC).



A variety of RGB combinations have been tested. FCC14 however, appeared to be the most
appropriate one for photolineaments discrimination. PC1 accounted for topography (essential
element in photointerpretation) whereas R5 reflected the spectral characteristics of vegetation
amount (for Sentinel-2, B5 component of ratio R5). Unfortunately, despite the contribution of
Sentinel-2, B7 in the same ratio, stratigraphic contacts were not easily detected except for FCC14
for the area under investigation.

**3.2. GIS Lineaments Evaluation**

The height values were divided in 15 classes and then coded in colors in order to better highlight
the relief. The artificial illumination of DEM was from a Northwest direction (315°) and from
45° altitude. From statistical point of view, almost one third (1/3) of the area is characterized by
slopes less than 8.7° whilst only 4,8% accounts for slopes >43°. As far as the orientation of
slopes is concerned, very few surfaces have a northern-northeastern tendency. The scenery
denotes all kinds of aspect of slopes.
The geology of the area under investigation with the potential tectonics as resulted from the
integration of the enhanced satellite data and the spatial information. The corresponding area
show that most of the faults detected on FCC 14 (Figure 4) have been already mapped whereas
several more apply for field cross-checking. The integration of Remote Sensing and Geographic
Information Systems has proven to be a reliable. method of fault mapping to a satisfactory
degree. Only at one side of the mainstream, smaller tributaries develop, which is a very good
example of such a valley is presented while the relative displacement of beds at the sides of the
streamline, safely enhances the suspicion of underlying fracturing.



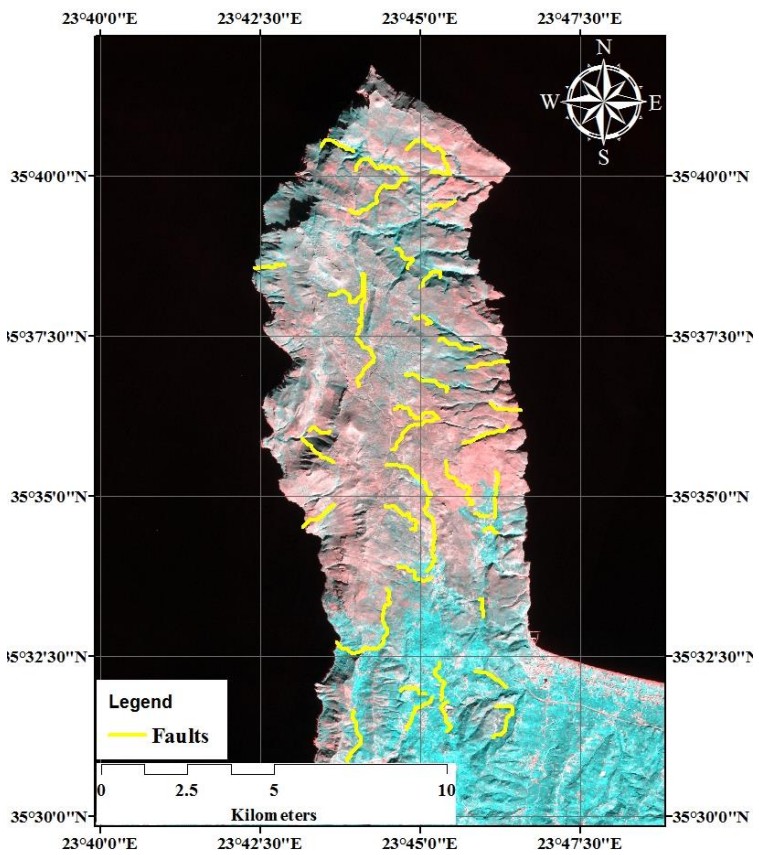


**Figure 4. Fault detection using False Color Composition FCC14**

Further extended fault zone recognized in the study area. It is the same feature observed in all
cases of digital image enhancement. The streamline following it, implies underlying fracturing.
A part of this zone has been already mapped. It is clear though, that the zone extends over a
larger area than the one found and mapped during field investigations.
Several more examples that enforced the belief of underlying failure as was initially suspected
during photointerpretation, and undoubtedly the drainage network has proven to be an
indispensable tool in the evaluation of photolineaments.



The final FCC14 that was used for photolineaments detection was composed of PC1, R5 and
Sentinel-2, B3 in Red, Green and Blue respectively. R5 on the other hand was a complex
function between Sentinel-2, B5 and Sentinel-2, B7. Sentinel-2, B5 is widely used for vegetation
amount investigations whilst Sentinel-2, B7 is very common in the field of Geology. Hence, the
variant tones of green in FCC14 would normally account for vegetation and/or geology. Since
the map of geology of the area was already in hand, in digital form no confusion was made. In
some cases, vegetation alignment could safely reveal underlying structural weakness whereas in
other cases, truncation and displacement of beds were sufficient for the evaluation of the
lineament concerned. Particular attention was paid when a particular lineament connected to a
geological boundary.
In order to evaluate the photolineaments detected on FCC14 as indicators of underlying
structural weakness, certain phenomena accompanying fracture zones had to be reckoned with.
Spatial models (GIS) enabled such estimations. Abrupt changes of slope and aspect, streamline
sudden bends and straight segments of streams were the criteria used in general terms.
Vegetation alignments and drag effects as were identified during photointerpretation, also
applied for a few more lineaments to be recognized. The results of faults detection on FCC14 by
Remote Sensing and GIS means whereas the differences between the results derived during this
work and conventional fault-mapping (during fieldwork) can be noticed.
**4. Conclusions**
Beyond a reasonable doubt, the integration of Remote Sensing and Geographic Information
Systems (GIS) in the · case of fault detection in designated study area gave satisfactory results. A
variety of enhancement techniques resulted in the discrimination of several photolineaments. Not
all of them related to fracturing. Cultural features such as roads were immediately extracted so

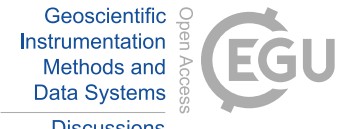

that no confusion would be made with faults. The same was true for drainage lines and
geological boundaries, which in many cases were strongly emphasized and could easily result in
false identification.
The best combination has proven to be for the first Principal Component (PC1 in red color-gun,
accounting for 82.27% of total scene variance), ratio product R5 = ((Sentinel-2, B5)$^2$ +( Sentinel-
2, B7)$^2$)$^{1/2}$, and one of the original bands (Sentinel-2, B3). For Sentinel-2, B5's contribution in
R5, the spectral characteristics of vegetation amount were emphasized (vegetation is particularly
responsive in Sentinel-2, B5). Hence, the preferential growth of vegetation along linear features,
suggested in many cases underlying fracturing. Despite Sentinel-2, B7's contribution in the same
ratio, the spectral responses of geological formations were not highlighted except for very few
cases of apparent displacement of beds. PC1 on the other hand accounted for the topographic
information that was lost due to ratio's contribution.
Spatial modeling (DEM and products) was crucial for the evaluation of photolineaments detected
on FCC14 (PC1, R5, in Sentinel-2, B3). The Digital Elevation Model of the area under
investigation and the derived maps of slope and aspect enabled several times to imply for
fracturing (wherever abrupt changes of slope and aspect were observed at the sides of a
particular photolineament).
The drainage network has proven to be particularly helpful and informative on the underlying
structures. Rift valleys and displacement of rivers' routes were common phenomena within the
scene of observation. In many cases parallel vegetation alignments differing from the vegetation
outcrop of the surrounding area (abrupt tonal differences) further assured us about possible
failure of the crust. Of course, the analysis was not based on the number of criteria satisfied each





time. A fault associated with more criteria than another does not necessarily make it a fracture
zone of more significance.
In comparison with the conventional methods of mapping (field investigations) most of the faults
recognized by automated watershed delineation, have already been mapped during field
investigations. Several of the zones found, were not discerned on the FCC14. That was due to the
following parameters that someone needs to bear always in mind before proceeding with
analogous to the present researches.
Faults mapping by using satellite data integrated with spatial information (GIS), may lead to
quite noteworthy results. Experience is crucial for an accurate photointerpretation.  In this case
study, most of the faults that have been mapped in the past during fieldwork investigations were
also identified on the satellite image. Some of the photolineaments discerned but were not
readily seen on FCC14. That should be for all factors explained earlier on.  Lastly, for all
photolineaments that were identified on the satellite image and raised suspicion of failure of the
crust, in- situ data collection for verification of the results is strongly recommended, in order to
be total aware of the tectonics of the area.

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
