# Peer review of "Integration of Remote Sense and Geographic Information Systems in Geological Faults Detection in Crete Island, Greece. Mohamed Elhag1\*, Dalal Alshamsi2 1 Department of Hydrology and Water Resources Management, Faculty of Meteorology, Environment & Arid Land Agriculture, King Abdulaziz University Jeddah, 21589. Saudi Arabia. 2 Department of Geology, College of Science, United Arab Emirates Univers"

_Geoscientific Instrumentation, Methods and Data Systems, 2018_

## Referee Comment (RC1) · N. Yilmaz (Referee) · 26 Oct 2018

Dear Author,

In my opinion your manuscript has a scientific quality. Your article needs only a few corrections which are below:

1. In the references section,

* In lines 346,369, 370,373, 376, 378, 384, 397, 402, 405, 408, 411, 413, 414, 416, 419 and 421 the authors name were written entirely open, please write the names in the correct style for the references section.

[Figure]

* In line 368 the author name was written entirely open, please write the name in the correct style for the references section. And add the journal name please.

2. In line 104, Elhag and Bahrawi,2016 or 2014? please check the year correctly.

3. Please add more information about the climate of the study area. For example the minimum and maximum temperatures according to the seasons and precipitation rates.

Best regards,

Nese YILMAZ

---

## Referee Comment (RC2) · G. Alexandrakis (Referee) · 30 Oct 2018

Dear author and Editor,

I have found the method interesting but there are much that is needed to be improved since the findings does not agree with the actual geological frame of the area. More over there some other things that you need to consider. In the abstract you are mentioning the Anatolian fault zone, which is not near the Study area. Even if the Anatolian fault is a large and important fault, has not a significant connection with the Cretan tectonic system which is influenced by the subduction zone of the European and African plate In the study area section (2.1) lines 110-112, I have not understand if you are

refer to the Aegean or the study area. But if you are referring to the study area the geology of the Kolympari area is known. The geological formations in the study area are Cretaceous limestones with some dolomites. There are some Miocene deposits and some alluvial sediments in the North. From the official IGME maps scale 1:50000 (Platanias & Kasteli) there are 5 major faults all in the N-S direction, which have been validated by field work in the past.

But the major issue are the results of this work. I am not making any comments on the method since the results are not close to the real situation. In line 263-264 you are mentioning that the results correspond to field cross checking but there is no reference to support this. Additionally in Mountrakis et al 2013 Neotectonic analysis, active stress field and active faults seismic hazard assessment in Western Crete, Bulletin of the Geological Society of Greece Vol. 47, 2013, there are no indications of those. Even though some of them are some parts of the mapped and validated from field work fault zones, but most of the structures that were identified are parts of the drainage network and karstic formations. More over the majority of the large faults zones in the area where not identified by this method.

For those reasons I have to reject this work.

Yours faithfully George Alexandrakis

---

## Referee Comment (RC3) · R. Castaldo (Referee) · 11 Nov 2018

Dear authors, the idea on which the manuscript is based is interesting and can be followed, but I think that the work, as is presented, reveals immature. For my point of view the manuscript has a lot of dark sides. I try to summarize them as questions:

1)why do you mention the anatolian fault zone? i think that there isn't direct relationship with the study area; 2)what santinel2 data do you use? a single or multiple images? of which period? No information about this aspect are treated. 3)One of your aim is the fault detection. Attention! not all the fractures or surface failures are faults..the fault implies a kinematic! 4)What is your integration criterion? it isn't clear how you integrate

the satellite data and the GIS information; 5)what kind of GIS layers do you use? I suggest you to consider also the structural map of the study area. It seems that not all the knowed faults have been individuated.

All these points represent critical aspects that aren't discussed. For these motivation i suggest to reject the manuscript in the present form encouraging the author to deeply work on this idea. best regards,

---

## Author Comment (AC1) · 9 Dec 2018

thanks for your effort your remarks will be considered in the final version yours
* * *

---

## Author Comment (AC2) · 9 Dec 2018

thanks for your effort your remarks will be considered in the final version yours

---

## Author Comment (AC3) · 9 Dec 2018

thanks for your effort your remarks will be considered in the final version yours
* * *